

# Control Simulation Experiments of Extreme Events with the Lorenz-96 Model

Qiwen Sun[1,2], Takemasa Miyoshi[1,3,4], and Serge Richard[1,2]

[1]Data Assimilation Research Team, RIKEN Center for Computational Science (R-CCS), Kobe, 650-0047, Japan
[2]Graduate School of Mathematics, Nagoya University, Nagoya, 464-8601, Japan
[3]Prediction Science Laboratory, RIKEN Cluster for Pioneering Research, Kobe, 650-0047, Japan
[4]RIKEN Interdisciplinary Theoretical and Mathematical Sciences Program (iTHEMS), Kobe, 650-0047, Japan

**Correspondence:** Qiwen Sun (qiwen.sun@riken.jp), Takemasa Miyoshi (takemasa.miyoshi@riken.jp)

**Abstract.** Control Simulation Experiment (CSE) is a recently developed approach to investigate the controllability of dynamical systems, extending the well-known Observing Systems Simulation Experiment (OSSE) in meteorology. For effective control of chaotic dynamical systems, it is essential to exploit the high sensitivity to initial conditions for dragging a system away from an undesired regime by applying minimal perturbations. In this study, we design a CSE for reducing the number
of extreme events in the Lorenz-96 model. The 40 variables of this model represent idealized meteorological quantities evenly distributed on a latitude circle. The reduction of occurrence of extreme events over 100 years runs of the model is discussed as a function of the parameters of the CSE: the ensemble forecast length for detecting extreme events in advance, the magnitude and localization of the perturbations, and the quality and coverage of the observations. The design of the CSE is aimed at reducing weather extremes when applied to more realistic weather prediction models.

**1  Introduction**

In the recent study Miyoshi and Sun (2022) of the first two authors, it has been demonstrated that a Control Simulation Experiment (CSE) leads to a successful control of the chaotic behavior of the famous Lorenz-63 model (Lorenz, 1963). Namely, by applying small perturbations to its nature run, the evolution of the system can be confined to one of the two wings of the attractor. The method is based on an extension of the Observing Systems Simulation Experiment (OSSE) from predictability
to controllability. This work can also be seen as an application of the control of chaos (Boccaletti et al., 2000) to the framework of Numerical Weather Prediction (NWP).

The control of weather is certainly one of the oldest dreams of human being, and several earlier works have already investigated different approaches of weather control based on numerical simulations. As for some of the pioneering works, let us mention Hoffman (2002) and Hoffman (2004) in which Hoffman discusses a general scheme and some applications for the
control of hurricanes. A case study is then presented in Henderson et al. (2005), where the temperature increments needed to limit the wind damage caused by Hurricane Andrew in 1992 is studied with a four-dimensional variational data assimilation technique (4D-Var). Such investigations have recently been further extended in Wang et al. (2021) for the control of cyclones, with some numerical simulation experiments based on the Typhoon Mitag of 2019. Similarly, in MacMynowski (2010) the





effect of small control inputs are evaluated by using the Cane-Zebiak 33 000-state model of the El-Niño/Southern Oscillation
(ENSO), and the outcome is a (theoretical) significant reduction of the ENSO amplitude with small control inputs. As a final
example, let us mention that weather and climate modifications can also be explored as a problem of optimal control, see for
example Soldatenko and Yusupov (2021).

In the present paper, the CSE proposed in Miyoshi and Sun (2022) is further developed and applied to the Lorenz-96 model
with 40 variables (Lorenz, 1995). In this setting, a notion of extreme events is defined first, based on the observation of 100
years of the nature run. Next, the aim of the CSE is to drive any new nature run away from the extreme events, by a suitable
application of small perturbations to the nature run whenever such an event is forecasted. By defining a success rate for such
actions, it is possible to assess the performance of the CSE as a function of the different parameters involved in the process.

The underlying motivation for such investigations is quite clear and can be summarized in a single question: Can one control
to avoid weather extremes within its intrinsic variability? Indeed, the sensitivity to initial conditions of the weather system
advocates the implementation of suitable small perturbations for dragging the system away from any catastrophic event. With
such an approach, heavy geoengineering actions become automatically obsolete. However, ahead of any such implementations,
numerical investigations have to demonstrate the applicability and the effectiveness of such actions, and ethical, legal, and social
issues have to be thoroughly discussed. With this ambitious program in mind, this paper provides a precise description of the
CSE applied to the Lorenz-96 model, and some sensitivity tests with the CSE parameter settings. Clearly, this corresponds to
a modest but necessary step before dealing with more realistic NWP models.

Let us now describe more concretely the CSE and the content of this paper. In Sect. 2 we briefly introduce the Lorenz-96
model (L96) with 40 variables, recall the main ideas of the Local Ensemble Transform Kalman Filter (LETKF), and describe
the implementation of the LETKF with 10 ensemble members for our investigations on L96. By creating synthetic observations
with the addition of uncorrelated random Gaussian noise to the nature run, we then test the performances of our system, and
get the analysis and forecast RMSEs for different forecast lengths. Multiplicative inflation and observation localization are also
briefly introduced in this section as essential techniques for stabilizing the LETKF.

In Sect. 3, we introduce the definition of extreme events for L96, and provide the precise description of the CSE. It essentially
consists in simulating the evolution of the system for a certain forecast length, integrating its evolution with the forth-order
Runge-Kutta scheme, and applying a suitable perturbation if an extreme event is forecast. The LETKF is then used for assimi-
lating the noisy observation to the forecast value in order to get the analysis value for the next integration step. Definitions for
a success rate and for a perturbation energy (energy delivered to the system through the perturbations) are also introduced in
this section, and a discussion of these quantities as functions of the forecast length and of the magnitude of the perturbation
vectors is provided. Based on a time scale commonly accepted for this model, our CSEs are taking place for a period of 100
years, the noisy observations are available every 6 hours, and the forecast length is chosen between 2 and 14 days.

Sensitivity tests are finally performed in Sect. 4. Namely, the performance of the method is evaluated when perturbation
vectors are applied at fewer places, or when the observations are collected less frequently. In these settings, the success rate
and the perturbation energy are again discussed as a function of the parameters. The reason for performing these tests is quite





clear: In the framework of a more realistic numerical weather prediction model, this would correspond to the restricted ability of applying a global perturbation (even of a small magnitude), or to lack of regular measurements of the weather system.

These investigations are only the second step towards the development of an applicable CSE, but the outcomes are already promising: For various settings a success rate close to 1 is obtained. Such a score depends on the forecast length, on the perturbation energy, on the number of sites on which the perturbations are applied, and on the regularity of the observations. The next step will be to consider a more realistic weather prediction model, and to get a more quantitative information about the exact amount of energy introduced in the system, and a precise evaluation of the false alarms. Indeed, the ultimate goal of

such investigations is to keep a high success rate, but to minimize the perturbation actions, in magnitude and in frequency. We plan to work on these issues in the future.

## 2    Background information

### 2.1    The Lorenz-96 model

Lorenz-96 model was first introduced in Lorenz (1995), and is expressed by a system of $K$ differential equations of the form:

$$\frac{dX_k}{dt} = (X_{k+1} - X_{k-2})X_{k-1} - X_k + F \tag{1}$$

for $k \in \{1, 2, \ldots, K\}$ and $K \geq 4$. Note that $F > 0$ is independent of $k$. It is also assumed that $X_0 = X_K$, $X_{-1} = X_{K-1}$, and $X_{K+1} = X_1$, which ensure that the system (1) is well-defined for each $k$. These conditions endow the system with a cyclic structure of the real *variables* $X_k$. Originally, these variables represented some meteorological quantities evenly-distributed on a latitude circle. In this framework, the constant term $F$ simulates an external forcing, while the linear terms and quadratic

terms simulate internal dissipation and advection respectively. A simple computation performed in (Lorenz and Emmanuel, 1998, Sect. 2) shows that the average value over all variables and over a long enough time lies in $[0, F]$, and that the standard deviation lies in $[0, F/2]$.

As already observed in (Lorenz, 1995, Sect. 2), when $F$ is small enough, all solutions decay to the steady solution defined by $X_k = F$ for $k = 1, 2, \ldots, K$. Then, as $F$ increases, the solutions show a periodic behavior, and end up with a chaotic behavior

for larger $F$. For the following discussion, we choose $K = 40$ and $F = 8$, which is identical to the setting used in Lorenz and Emmanuel (1998). This setting ensures the chaotic behavior of the model. By studying the error growth, namely the difference between two solutions with slightly different initial values, it has been found that the leading Lyapunov exponent corresponds to a doubling time of $0.42$ unit of time (Lorenz and Emmanuel, 1998, Sect. 3). By comparing this model with other up-to-date global circulation models, it is commonly accepted that $1$ unit of time in this model is equal to $5$ days in reality. As a

consequence, the doubling time is about 2.1 days. Note that the doubling time may decrease when $F$ increases further.

In (Lorenz and Emmanuel, 1998, Sect. 3), some numerical investigations are performed on this model. In particular, the evolution of one variable is reported in Fig. 4 of this reference over a long period of time, and this evolution is compared with the evolution of the same variable when a small perturbation is added at the initial time. For one month, the two trajectories are not distinguishable, while 2 months after the perturbation the two trajectories look completely independent. In the sequel, we shall




consider similar perturbations of the system, but applied to more than one variable. More precisely, since the evolution is taking place in $\mathbb{R}^{40}$, we shall consider perturbation vectors in this space, of different directions and of different sizes (magnitudes). By applying suitable perturbation vectors, our aim is to drive the evolution of the system in a prescribed direction within a limited time interval.

## 2.2    Local Ensemble Transform Kalman Filter

The *Local Ensemble Transform Kalman Filter* (LETKF) is a type of ensemble Kalman filter which was first introduced in Hunt et al. (2007). It corresponds to a data assimilation method suitable for large, spatiotemporally chaotic systems. For $i \in \{1, \ldots, N\}$, let $\mathbf{x}^{b(i)}$ denote a $m$-dimensional state vector, at a given fixed time. The exponent $b$ stands for *background*, or *forecast*. One aim of LETKF is to transform the ensemble $\{\mathbf{x}^{b(i)}\}_{i=1}^{N}$ into an *analysis* ensemble $\{\mathbf{x}^{a(i)}\}_{i=1}^{N}$ such that its analysis mean $\overline{\mathbf{x}^a}$ minimizes the cost function:

$$J(\mathbf{x}) = \left(\mathbf{x} - \overline{\mathbf{x}^b}\right)^T (\mathbf{P}^b)^{-1} \left(\mathbf{x} - \overline{\mathbf{x}^b}\right) + \left[\mathbf{y} - H(\mathbf{x})\right]^T \mathbf{R}^{-1} \left[\mathbf{y} - H(\mathbf{x})\right]. \tag{2}$$

In this expression, $\mathbf{P}^b$ denotes the background covariance matrix defined by $(N-1)^{-1}\mathbf{X}^b(\mathbf{X}^b)^T$, where $\mathbf{X}^b$ is the $m \times N$ background ensemble perturbation matrix whose $i^{\text{th}}$ column is given by $\mathbf{x}^{b(i)} - \overline{\mathbf{x}^b}$. $\mathbf{R}$ stands for the observation error covariance matrix and $H$ corresponds to the observation operator. For LETKF, it is assumed that the number of ensemble members $N$ is smaller than the dimension $m$ of the state vector, and also smaller than the number of observations (which corresponds to the dimension of $\mathbf{y}$). In Hunt et al. (2007), it is shown that the minimization can be performed in the subspace $S$ spanned by the

vectors $\mathbf{x}^{b(i)}$, which is efficient in terms of reduced dimensionality.

To perform the analysis, the matrix $\mathbf{X}^b$ is used as a linear transformation from some $N-$dimensional space $S'$ onto $S$. Thus, assume that $\mathbf{w} \in S'$, then $\mathbf{X}^b\mathbf{w} \in S$ and $\overline{\mathbf{x}^b} + \mathbf{X}^b\mathbf{w} \in S$ are vectors in the model variables. Similar to Eq. (2), a cost function $J'$ for $\mathbf{w}$ is defined by

$$J'(\mathbf{w}) = (k-1)\mathbf{w}^T\mathbf{w} + \left[\mathbf{y} - H\left(\overline{\mathbf{x}^b} + \mathbf{X}^b\mathbf{w}\right)\right]^T \mathbf{R}^{-1} \left[\mathbf{y} - H\left(\overline{\mathbf{x}^b} + \mathbf{X}^b\mathbf{w}\right)\right]. \tag{3}$$

In (Hunt et al., 2007, Sect. 2.3), it is shown that if $\overline{\mathbf{w}^a} \in S'$ minimizes $J'$, then the vector $\overline{\mathbf{x}^a} := \overline{\mathbf{x}^b} + \mathbf{X}^b\overline{\mathbf{w}^a}$ minimizes $J$.

Let us now define the vectors $\mathbf{y}^{b(i)} := H(\mathbf{x}^{b(i)})$ and the corresponding matrix $\mathbf{Y}^b$ whose $i^{\text{th}}$ column is given by $\mathbf{y}^{b(i)} - \overline{\mathbf{y}^b}$. The linear approximation of $H\left(\overline{\mathbf{x}^b} + \mathbf{X}^b\mathbf{w}\right)$ is then provided by the expression $\overline{\mathbf{y}^b} + \mathbf{Y}^b\mathbf{w}$. By inserting this approximation into Eq. (3), one obtains an expression having the form of the Kalman filter cost function, and by standard computations, the solution $\overline{\mathbf{w}^a}$

and its analysis covariance matrix $\widetilde{\mathbf{P}}^a$ can be inferred. By transforming this solution back to the model space, one gets

$$\overline{\mathbf{x}^a} = \overline{\mathbf{x}^b} + \mathbf{X}^b\overline{\mathbf{w}^a}, \tag{4}$$

$$\mathbf{P}^a = \mathbf{X}^b\widetilde{\mathbf{P}}^a(\mathbf{X}^b)^T. \tag{5}$$

Finally, a suitable analysis ensemble perturbation matrix $\mathbf{X}^a$ can be defined by $\mathbf{X}^a := \mathbf{X}^b\mathbf{W}^a$, where $\mathbf{W}^a := [(N-1)\widetilde{\mathbf{P}}^a]^{\frac{1}{2}}$. Here, the power $\frac{1}{2}$ represents the symmetric square root of the matrix. One then easily checks that the sum of columns of $\mathbf{X}^a$

is zero, and that the analysis ensemble $\mathbf{x}^{a(i)}$ (obtained by the sum of the $i^{\text{th}}$ column of $\mathbf{X}^a$ and $\overline{\mathbf{x}^a}$) agrees with the mean and covariance matrix given by Eq. (4) and Eq. (5).





## 2.3 Application of LETKF to the Lorenz-96 model

From now on, we consider the Lorenz-96 model with $40$ variables and $F = 8$. The vector generated by the $40$ variables will be used as the state vector $\mathbf{x}$ for the LETKF, and will be referred to as the state variables. To study the predictability and

analysis accuracy of LETKF, we run the model by using a forth-order Runge-Kutta scheme. The initial value is provided in the Appendix, and the integration time step is $0.01$ unit of time. As mentioned in Sect. 2.1, $1$ unit of time is assumed to equal to $5$ days in reality. We let the model evolve for $110$ years ($8030$ units of time) and disregard the first $10$ years' results to avoid any transient effect. Note that for all subsequent experiments, the time $0$ is set after these disregarded $10$ years. For $100$ years of nature run, we generate a synthetic observation every $6$ hours ($0.05$ unit of time) by adding independently a $Normal(0,1)$

distributed random number to the value of each variable. For the choice of the amplitude of these perturbations, we recall that for the Lorenz-96 model with $F = 8$, the average value over all variables and over a long enough time lies in $[0, 8]$, and the standard deviation lies in $[0, 4]$.

To proceed with data assimilation, we consider $10$ ensemble members. For the initial data, we add to the nature run at time $0$ a $Normal(0,1)$ distributed random number independently to each variable. We then run the forecast-observe-analyze cycle

every $6$ hours over $100$ years with LETKF and with the synthetic observations. To avoid underestimating the error variance, we implement a *multiplicative inflation* with $\rho = 1.06$, namely we consider an inflated background error covariance matrix $\rho \mathbf{P}^b$ instead of $\mathbf{P}^b$, see (Houtekamer and Zhang, 2016, Sect. 4). Also, with the current parameter's setting of the Lorenz-96 model, the attractor has $13$ positive Lyapunov exponents (Lorenz and Emmanuel, 1998, Sect. 3) which is greater than the number of ensemble members. In this case, the forecast errors will not be corrected by the analysis, because the errors will

grow in directions which are not accounted for by the ensemble (Hunt et al., 2007, Sect. 2.2.3). This problem can be solved by implementing a *localization*, namely by performing the analysis independently for each state variable and by taking the distance between variables into account.

In this study, we implement the $\mathbf{R}$ localization described in (Hunt et al., 2007, Sect. 2.3). More precisely, every $6$ hours we update each state variable independently, which means that we perform the analysis $40$ times, and each time we update one state

by using its forecast ensemble and some truncated observations. For the space dependence, recall that the states are located on a latitude circle, and therefore the analysis result of a given state should be affected more by its neighboring states rather than by states which are located farther from it. To implement this idea, let us consider how we perform of the analysis of the variable $i$. We first fix a suitable diagonal observation error covariance matrix $\mathbf{R}$ which takes into account the distance between the site $i$ and any site $j$: the entry $(j,j)$ of this matrix is given by $e^{d(i,j)^2/2L^2}$, where $d(i,j) = \min\left\{|i - j \bmod 40|, |j - i \bmod 40|\right\}$ if

$d(i,j) \leq 19$, and $L := 5.45$. Then the analysis of state $i$ is performed with the formulas of Sect. 2.2 on a space of dimension $39$, which means that the state and the observations associated with $j = i + 20 \bmod 40$ are disregarded. Note that the value of the tuning parameter $L$ (and the related choice of a truncation at the maximum distance of $19$) have been fixed by a minimization process of the analysis RMSE (obtained by the difference between the analysis values and the nature run). In this setting, which will be kept throughout the investigations, the analysis RMSE is $0.1989$.


We also test the forecast RMSE for different forecast lengths. To do this, we run the forecast with the analysis value as initial value, and compute the RMSE by comparing the forecast with the nature run at different time points. Note that the previous experiment supplies a new initial analysis value every 6 hours. Figure 1 summarizes the forecast ability by providing the values of the forecast RMSE for different forecast lengths.

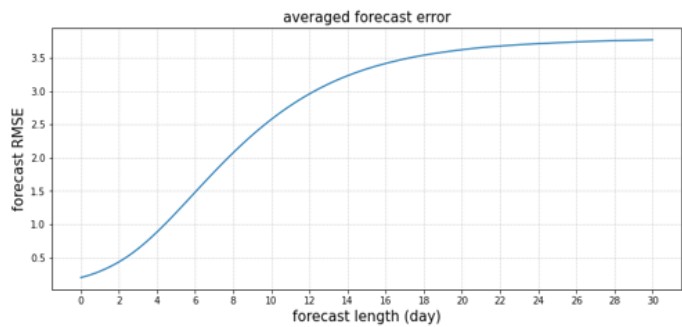

**Figure 1.** The forecast RMSE averaged over every 6 hours in 100 years run

## 3 The Control Simulation Experiment

In this section, we explain the design and the settings of the Control Simulation Experiment (CSE). Several results are already presented in this section, while additional discussion are gathered in the following section. The aim of this CSE is to avoid extreme events, and this will be achieved by adding small perturbations to the dynamical system. These perturbations should drive the system into a preferable direction. Let us mention that some preliminary experiments consisting in replacing the perturbation vectors by randomly generated vectors have shown the same distribution of state values as the nature run without 165 control. Namely, by applying random perturbations one can not reduce the number of extreme events.

### 3.1 Extreme events

Let us first provide a definition of the extreme events for the Lorenz-96 model. For that purpose, we use the nature run mentioned in Sect. 2.3, namely the trajectories of the 40 state variables recorded every 0.01 unit of time and over 7300 units (100 years). In each period of 0.05 unit of time (6 hours), we look at the maximum value over all states, and keep this value as 170 the local maximum for this period. The first 200 greatest local maxima over the 100 years are treated as extreme values. Note that the interest of considering intervals of 6 hours is to minimize the representation of events involving several variables taking large values simultaneously or within a very short period of time. As a result of this definition, extreme values appear 2 times a year on average. This construction leads also to a lower threshold value for an extreme event at 14.217. Figure 2a shows the histogram of the state values (nature run) recorded every 0.01 unit of time for 7300 units, and Fig. 2b shows the value of all 175 states over one randomly chosen year. Values that exceed the threshold are colored in black.


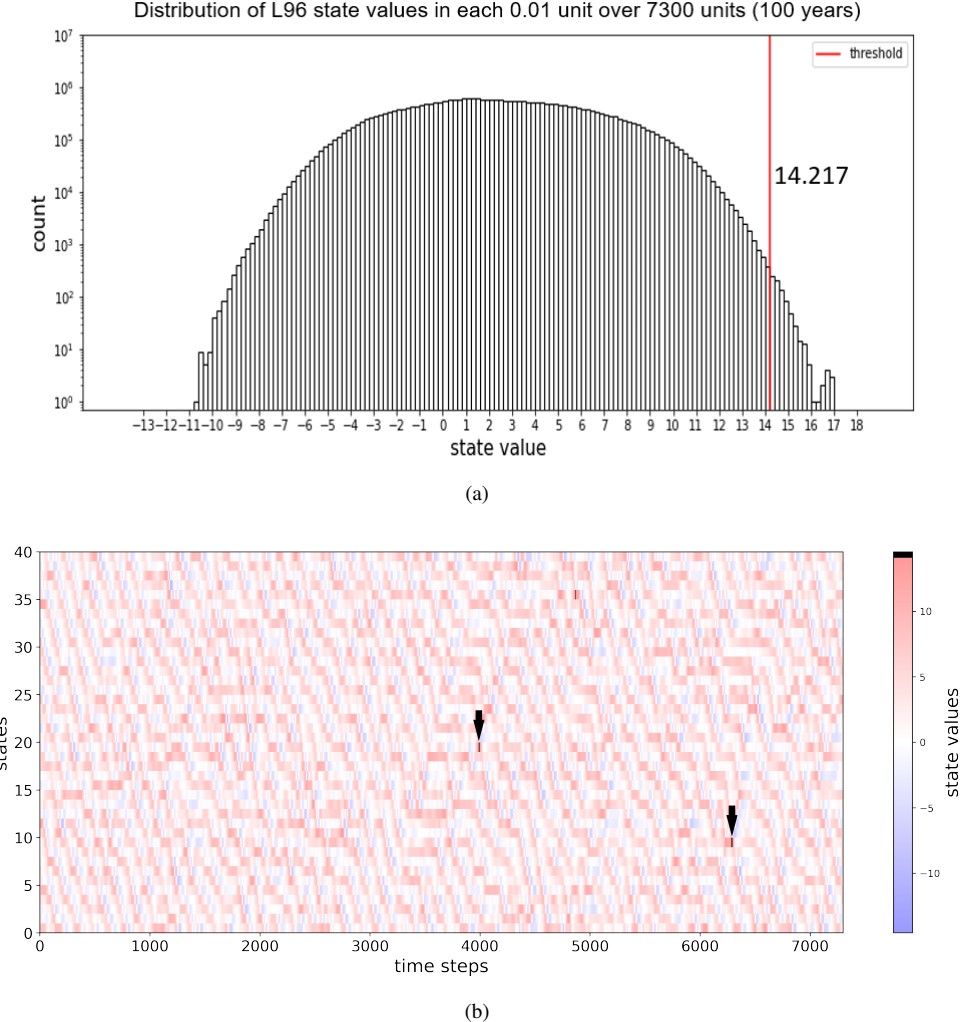

(a)

(b)

**Figure 2.** (a) The distribution of the state values of Lorenz-96 recorded every 0.01 unit of time for 7300 units, (b) The values of all states over a randomly chosen year. The extreme values are colored in black and indicated by arrows.





## 3.2 The Control Simulation Experiments

Recall that synthetic observations are produced every $0.05$ unit of time (6 hours). For the forecasts described below, a forecast length $T$ equal to $2$, $4$, $6$, $10$, or $14$ days is chosen.

1. At a certain time point $t_m$, we run a $T$ days forecast by integrating the Lorenz-96 model with initial values provided by the analysis ensemble value at time $t_m$. The integration is performed by using a forth-order Runge-Kutta scheme with integration step of $0.01$ unit of time.

2. If no extreme event is forecast during these $T$ days, then we just move to Step 3. If the maximum among all states exceeds the threshold at any time point during these $T$ days, then we apply some perturbations to the nature run from time $t_m + 0.01$ to time $t_m + 0.04$. After each perturbation, we use the perturbed system as initial value for the next integration over $0.01$ unit of time. After the fourth perturbation, the system evolves to $t_m + 0.05$.

3. At time $t_m + 0.05$, we have the forecast ensemble and the observation generated by the independent addition of a $Normal(0,1)$ distributed random number to each variable of the nature run obtained from Step 2. The LETKF is used to assimilate the observation, and an analysis ensemble at time $t_m + 0.05$ is produced. We then move back to Step 1 with $t_m$ replaced by $t_m + 0.05$.

Figure 3 shows the steps which are described above. In the upper part of the figure, the green curves represent the $T$ days forecast of one ensemble member. When a green curve crosses or touches the gray line representing the threshold value, it means that one (or more) state has a forecast value which is greater than or equal to the threshold. In the lower part of the figure, the black curves represent the nature run: the original nature run for the dashed curve, and the perturbed nature run for the solid curve. We shall call *the controlled nature run* the nature run obtained by the above process, and emphasize that it corresponds to a nature run including the perturbations, when these ones are applied.

Let us now describe how the perturbation vectors are chosen. For their definition we use two ensemble members $A$ and $B$ among the $10$. The ensemble $A$ corresponds to the ensemble showing the maximal extreme value during the forecast $T$ days. The exact place (one variable $k$) where this maximum is reached, and its precise time $t$ are recorded . For the ensemble member $B$, we first eliminate the candidates which show an extreme value at any location and at any time within the $T$ days. With the remaining members, we choose the member who has the minimal value for the variable $k$ and at the time $t$, see Fig. 4. Then, a rescaled difference of the ensemble members $B$ and $A$ at times $t_m + 0.01$ to $t_m + 0.04$ is applied to the nature run at times $t_m + 0.01$ to $t_m + 0.04$, as indicated in the above Step 2. Let us emphasize that the perturbation vectors are $40$ dimensional vectors, since the difference is computed on all variables. For the (Euclidean) norm of the perturbation vectors we choose them proportional to the value of the analysis RMSE $D_0 := 0.1989$. Thus, we fix the norm of this vector as $D := \alpha D_0$ with the magnitude coefficient $\alpha = 0.1, 0.2, \ldots, 1.0$. Since the norm of the average displacement in each $0.01$ unit of time is $1.1720$, the norm of the perturbations correspond to less than $2\%$ of the average displacement when $\alpha = 0.1$ and to $17\%$ of the average displacement when $\alpha = 1.0$.


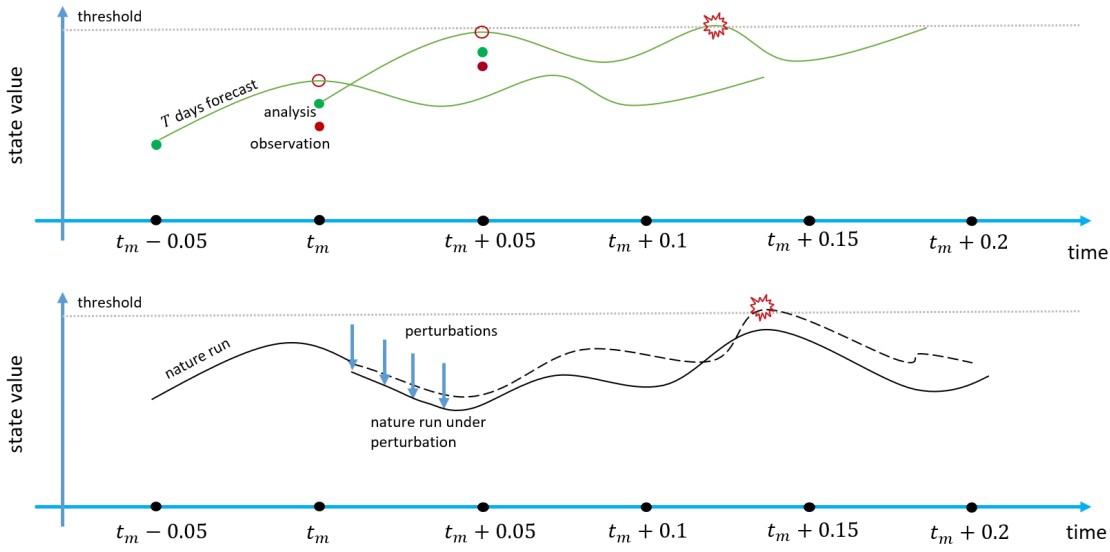

**Figure 3.** CSE description: When a $T$ days forecast from $t_m$ shows extreme values, perturbations are applied to the nature run between $t_m$ and $t_m + 0.05$. At time $t_m + 0.05$, previous forecast and observation based on the controlled nature run are available, and a new analysis ensemble is created.

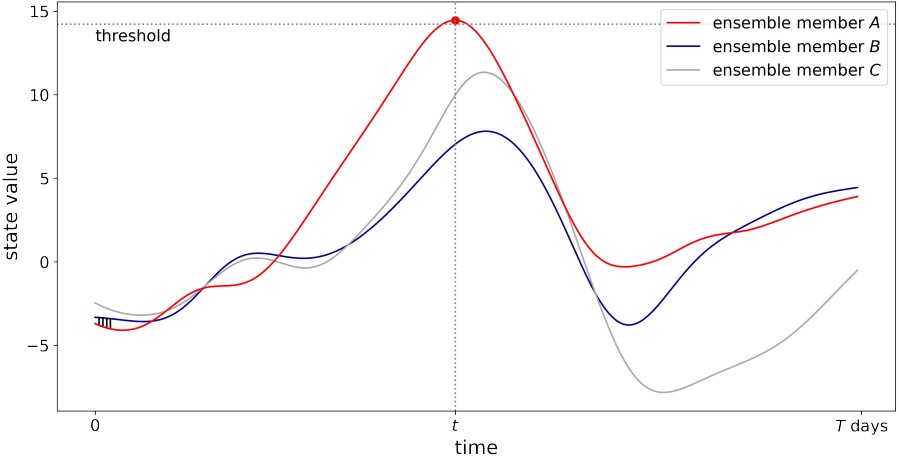

**Figure 4.** The forecast of a state with three ensemble members. Ensemble $B$ is chosen with the farthest distance to ensemble $A$ when $A$ exhibits an extreme event. The four perturbation vectors (before rescaling) are indicated on the lower left part of the figure.





If ever the above process does not allow us to choose an ensemble member $B$, then we come back 6 hours before, use the analysis ensemble at time $t_m - 0.05$, and run a $T$ days plus 6 hours forecast to find a suitable candidate.

In Fig. 5, we superpose the distribution of the state values recorded every 0.01 unit of time for 7300 units already obtained in Fig. 2a with the similar distribution for the controlled nature run. The parameters chosen are $T = 10$ days and $\alpha = 0.2$. It is clear that the number of values that exceed the threshold decreases significantly. The successive perturbations that were applied to the nature run effectively control the system and reduce the number of extreme events.

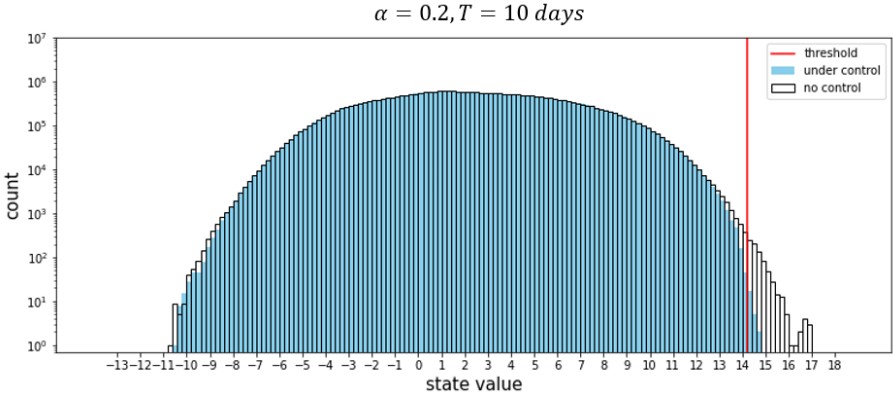

**Figure 5.** Distribution of state values for the nature run and for the controlled nature run.

When the norm of the perturbation vector is too small, namely when $\alpha$ is too small, the control is less effective for avoiding extreme events. Similarly, when the forecast length is too short, it is often too late for applying successfully the perturbations. In order to understand the effectiveness of the choice of $\alpha$ and $T$, we perform the same experiment for different combinations of these parameters. For each combination, we run 10 independent experiments. To be consistent with the definition of extreme events, we define the effectiveness as the success rate given by the formula

$$\text{success rate} := 1 - \frac{\#\{6 \text{ hours intervals in 100 years with extreme events}\}}{200}.$$

The outcomes of these investigations are gathered in Fig. 6.

In addition to the success rate, we also would like to know how many times a control of the system has been implemented, or equivalently how many times perturbation vectors have been applied. For that purpose, we call *a perturbation action* each group of four perturbations, as described in the above Step 2. We provide in Fig. 7a the averaged values of perturbation actions over a period of 100 years and obtained with 10 independent simulations. The intervals corresponding to one standard deviation are also indicated. However, since the magnitude of the perturbations vectors are proportional to $\alpha$, another quantity of interest is the *the perturbation energy* delivered to the system and defined by

$$\text{perturbation energy} := 4\alpha D_0 \times \#(\text{perturbation actions}).$$

Figure 7b provides the information about the perturbation energy over a period of 100 years. In other words, this value describes the total energy put into the dynamical system for controlling it during the 100 years run.

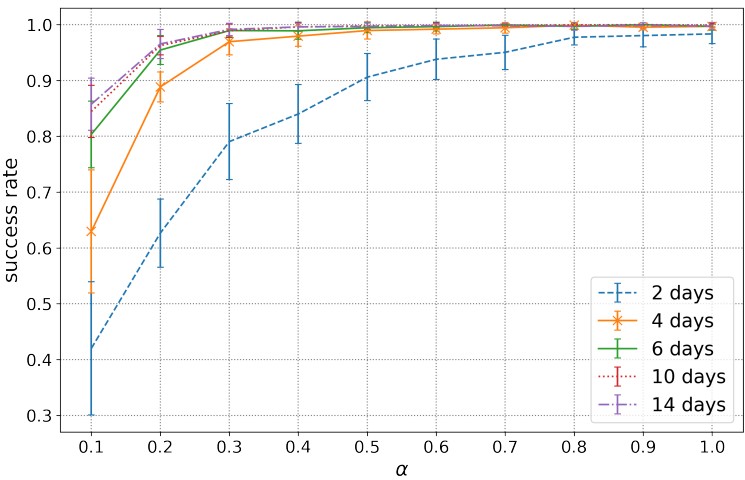

**Figure 6.** Success rates, as a function of the forecast length $T$ and of the magnitude coefficient $\alpha$.

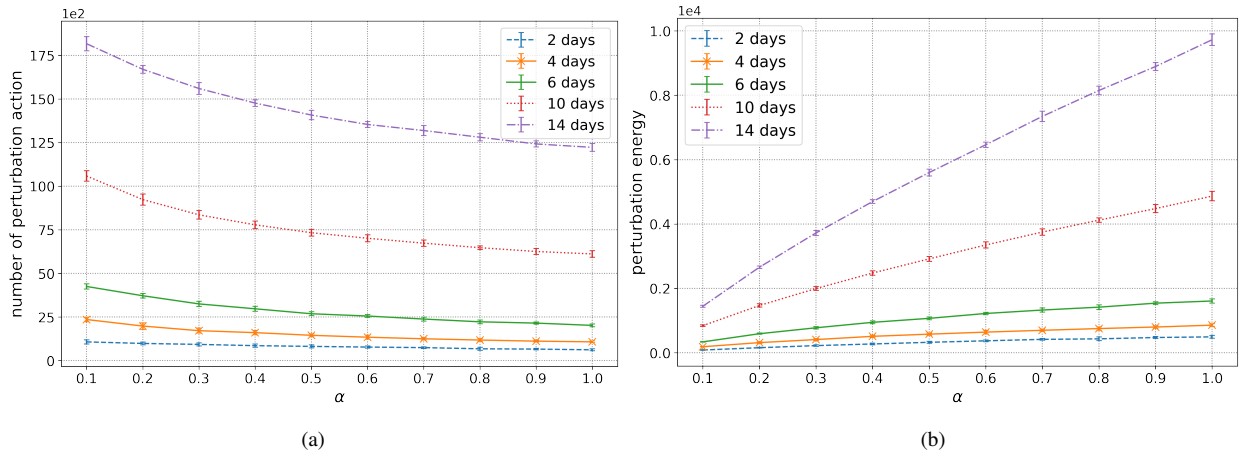

(a)                                                                                 (b)

**Figure 7.** Number of perturbation action and perturbation energy over a period of 100 years, as functions of the forecast length $T$ and of the magnitude coefficient $\alpha$.





By looking at Fig. 6, we observe that the success rate is quite low for the combination of short forecast length $T$ and small coefficient $\alpha$. Nevertheless, the lowest success rate is between $0.4$ and $0.5$, which means that more than $40\%$ of the extreme events are successfully removed. With $T$ longer than or equal to $4$ days and $\alpha$ greater than $0.5$, all performances

are quite similar: a success rate close to $1$. On the other hand, it is clearly visible on Fig. 7b that the perturbation energy quickly increases with longer forecasts and larger coefficients $\alpha$. In fact, the relation between the perturbation energy and the parameters $T$ and $\alpha$ is quite subtle. On the one hand, one observes that for constant $T$, the number of perturbation actions is decreasing as a function of $\alpha$, see Fig. 7a. Nevertheless, this decay is counterbalanced by the multiplication by the factor $\alpha$ itself, when computing the perturbation energy for any fixed $T$, as it appears in Fig. 7b. On the other hand, the increase of $T$,

for any fixed $\alpha$, has a simple impact in both Figures 7a and 7b: the corresponding values increase.

The increases with respect to $T$ are quite clear: A longer forecast increases the probability of having one particle exhibiting an extreme value, and thus increases the number of perturbation actions triggered. This also result in an increase of the perturbation energy. For the dependence on $\alpha$: a small value of this parameter will not be sufficient for dragging the system far away from an extreme event, and therefore a particle has a bigger chance to forecast an extreme event even after a perturbation action.

On the other hand, a large value of $\alpha$ corresponds to larger perturbation vectors moving the system momentarily away from an extreme event. In summary and according to these observations, for a pretty good success rate with a small energy cost, the choice $T = 4$ days and $\alpha = 0.5$ looks like an efficient compromise.

## 4   Local perturbations and partial observations

In the previous section, information was available for the $40$ variables, and the perturbation vectors were applied at the $40$

locations. We shall now discuss the situation when only partial information is available, and when the perturbation vectors are applied only at a reduced number of sites.

### 4.1   Local perturbations

The reason of considering local perturbations is quite straightforward: Since the $40$ variables of the Lorenz-96 model represent meteorological quantities evenly distributed on a latitude circle, it could be that perturbations can only be generated at a limited

number of locations. Also, one may be interested in using only a smaller number of sites, in order to reduce the perturbation energy delivered to the system. The purpose of this section is to test if such an approach is possible and efficient.

To study the interest and feasibility of local perturbations, we design two experiments: In the first one, we assume that a randomly selected set of locations can generate perturbations. During one realization (a 100-year run), these locations are fixed. In the second one, we assume that each location can generate perturbations, but only sites which are close to the targeted

site, namely the site corresponding to the state variable at which an extreme event is forecast, will be turned on and generate perturbations. Quite naturally, we shall refer to the first experiment as the *random sites* experiment, while the second one will be referred to as the *neighbor sites* experiment.





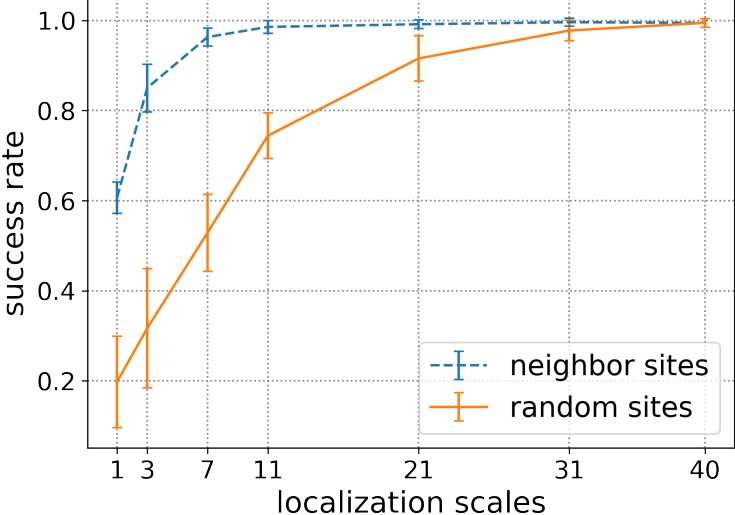

**Figure 8.** Success rates for the two experiments and for different localization scales. The forecast time $T$ is set to 4 days and the magnitude coefficient $\alpha$ to 0.7.

Recall now that the perturbation vector was previously obtained by the rescaled difference of two ensemble members. For local perturbations, we shall additionally set some entries of the perturbation vector to 0. In other words, some locations among the 40 states will not be perturbed, even if a perturbation is applied to other states. These latter sites will be called *the perturbation sites*, but observe that the perturbation can accidentally be also equal to 0 at some of these sites. For the random sites experiments, we choose successively the numbers of perturbation sites as 1, 3, 7, 11, 21, and 31. These numbers are referred to as *the localization scales*. For the neighbor sites experiments, we shall successively consider perturbation sites which are located at a distance at most 0, 1, 3, 5, 10, and 15 from the targeted site. According to this maximal distance, the localization scale will also coincide with 1, 3, 7, 11, 21, and 31. For both experiments and when an action is suggested by the forecast, we first compute the difference of two the ensemble members are mentioned above, we rescale the resulting vector with the coefficient $\alpha = 0.7$, and set to 0 its entries which are not at a perturbation site. This final vector is used for the perturbation, and its Euclidean norm provides the energy of this perturbation. For the perturbation energy, we consider the sum of these norms.

In Fig. 8, we provide the success rate for the different localization scales: 1, 3, 7, 11, 21, 31, and 40. The coefficient $\alpha$ is fixed to 0.7, and the forecast length $T$ to 4 days. The error bars mark the range within one standard deviation of 10 independent realizations. Clearly, using random sites is less effective than using neighbor sites. For localization scale equal to 1, which means only one site is perturbed, localizing the perturbation on the targeted site eliminates already more than 60% of the extreme events, while choosing a random position for the perturbation only eliminates about 20% of the extreme events. When the localization scale increases, the success rates of both experiments increases. However, when the localization scale is greater than 7, the success rates of the neighbor sites experiment does not increase anymore, while there is a constant increase for the





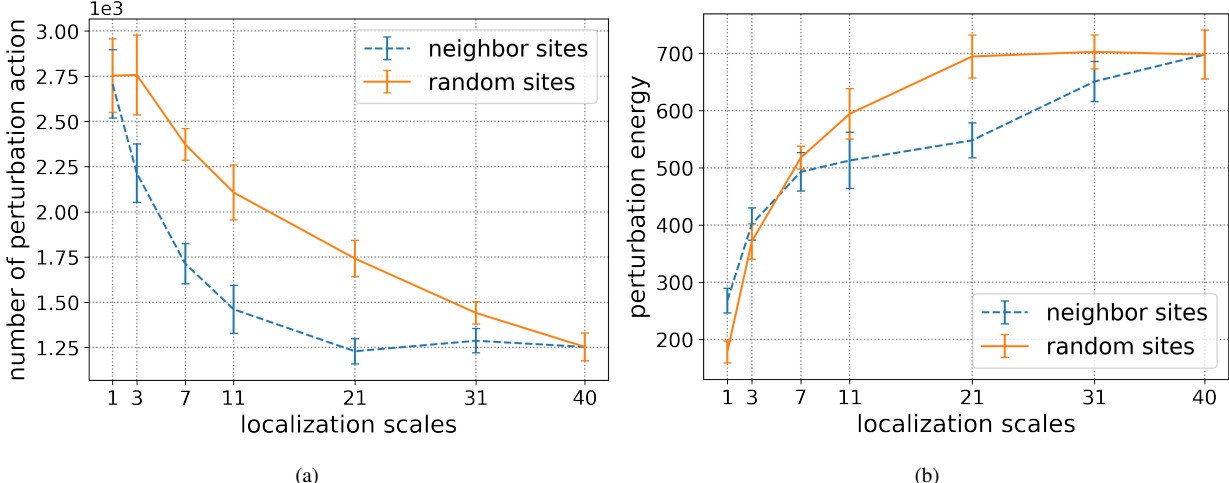

(a)                                               (b)

**Figure 9.** Number of perturbation action and perturbation energy for the two experiments and for different localization scales. The forecast time $T$ is set to $4$ days and the magnitude coefficient $\alpha$ to $0.7$.

random sites experiment, with a maximum at the localization scale equal to $40$. For the uncertainty range, the neighbor sites experiment also shows a more stable performance compared to the random sites experiment.

At the level of the number of perturbation action, see Fig. 9a, the neighbor sites experiments require less perturbation actions than the random sites experiments. In addition, when the localization scale is greater than $21$, the number of perturbation action for the neighbor sites experiments does not decrease anymore. This suggests the existence of an optimal setting for the localization scale around $21$. One the other hand, in Fig. 9b, the relation between the two experiments is a little bit more confused, but for a localization scale bigger than or equal to $7$, it also clearly appears that the neighbor sites experiment is more effective than the random sites experiment. For these localization scales, the perturbation energy is higher for the random sites experiment, even though the corresponding success rates are always smaller. In fact, more perturbation actions are taking place in the random setting, since the system is not carefully driven away from a forecast extreme event. For the localization scale smaller than $7$, the small perturbation energy for the random site experiment is due to the random choice of the components of the difference between the two ensemble members: in general these components are not related to the variable of the extreme event, and therefore to the biggest difference between the components.

By looking simultaneously at Figures 8, 9a, and 9b, let us still mention an interesting outcome: For a localization scale equal to $11$, the success rate is equal to the success rate obtained for the localization scale equal to $40$, but the corresponding perturbation energy in the former case is about $\frac{6}{7}$ of the perturbation energy used for controlling the $40$ sites. This ratio can even decrease to about $\frac{5}{7}$ for a localization scale of $7$, if we accept a small decay of the success rate. Thus, with a fine tuning of the localization scale, a reduction of the total perturbation energy can be obtained, without any decay of the success rate.



## 4.2 Partial observations


Our next aim is to study the performance of the CSE when only partial observations are available. The lack of information can either be spatial (with the observation of only 20 state variables) or temporal (with observations available only every 24 hours, which means every $0.2$ unit of time). We provide the description of both experiments simultaneously.

When observations are sparse in space or in time, the available information from observations decreases, and the analysis becomes less accurate, accordingly. As a consequence, the parameter $\rho$ of the multiplicative inflation and the **R** localization parameter $L$, introduced in Sect. 2.3, need to be adapted. In particular, suitable values for these parameters can lead to a smaller analysis RMSE. Note that the average spread of ensemble members should be close to the RMSE, namely, the value of average should spread basically explain the potential accuracy of the analysis results. Recall that the spread is computed as the RMSE but by replacing the truth by the mean value of the ensemble members.


For the observations sparse in space, the 20 observed state variables are still chosen evenly distributed on a latitude circle. The average spread and RMSE are computed on a 100 years data assimilation experiment with different combinations of parameters $\rho$ and $L$, and the results are provided in Figures 10a and 10b. The smallest RMSE together with a spread of a similar value are found for the combination $(\rho, L) = (1.08, 4.85)$, and this common value is about $0.31$ (to be compared with the value $0.1989$ of the original experiment).


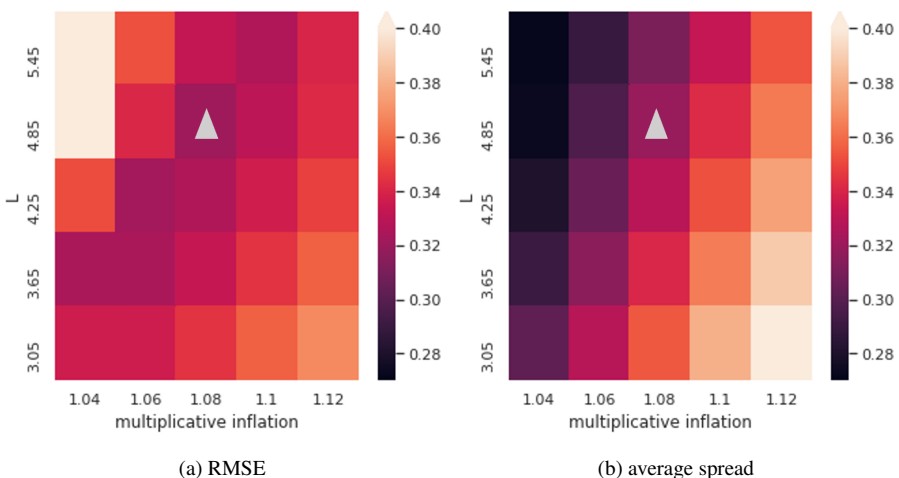

(a) RMSE                    (b) average spread

**Figure 10.** The RMSE (a) and the average spread (b) over 100 years run for different combination of multiplicative inflation parameter $\rho$ and **R** localization parameter $L$.


For the observations sparse in time, we have only tuned the parameter $\rho$ and kept the **R** localization parameter $L$ at its initial value, since the space structure has not changed. Table 1 provides the RMSE and the spread for different value of $\rho$ computed also on a 100 years data assimilation experiment, and the smallest RMSE is reached for $\rho = 1.50$, with a value almost two times bigger than the original RMSE.


| $\rho$ | 1.40 | 1.45 | **1.50** | 1.60 | 1.70 |
|---|---|---|---|---|---|
| RMSE | 0.443 | 0.429 | **0.422** | 0.427 | 0.434 |
| spread | 0.372 | 0.380 | **0.389** | 0.403 | 0.417 |

**Table 1.** The RMSE and spread over 100 years run with different values of $\rho$.

With the parameters mentioned above, we run 10 independent CSE for $T = 4$ days and $\alpha = 0.1$, $0.5$ and $1.0$. For the observations sparse in time, namely when an observation is taken place every 24 hours only, the system is still perturbed between two observation time and every every $0.01$ unit of time. More precisely, if the forecast computed from a time $t_m$ anticipate an extreme event, then the perturbation vectors are applied from time $t_m + 0.01$ to time $t_m + 0.19$, every $0.01$ unit of time.

The various results obtained for partial observations are gathered in Fig. 11. For a comparison, we also recall the results of the original experiment which assumes that all states are observed every 6 hours. Quite surprisingly, when half of the states are masked, and despite an increase of the analysis error by about $50\%$, the decay of the success rate is pretty small for $\alpha = 0.1$ and $0.5$, and negligible when $\alpha = 1.0$. On the other hand, the corresponding perturbation energy is higher than for the original setting, meaning that a more frequent application of the perturbation vectors is taking place. This might be caused by less accurate forecasts and by less efficient perturbations.

In contrast, when the observations are taking place every 24 hours only, the decay of the success rate is more clear, and the increase of the perturbation energy is obvious. For the success rate, the bigger RMSE can certainly explain why the analysis ensemble members can not make accurate enough forecasts even on a very short range (2 days). On the other hand, the increase of the perturbation energy is linked to the repeated application of the perturbation vectors, which took place 19 times instead of 4 times at every alerts. In summary, the partial forecasts, both in space and in time, lead to a CSE which is less efficient and necessitates more perturbation energy.

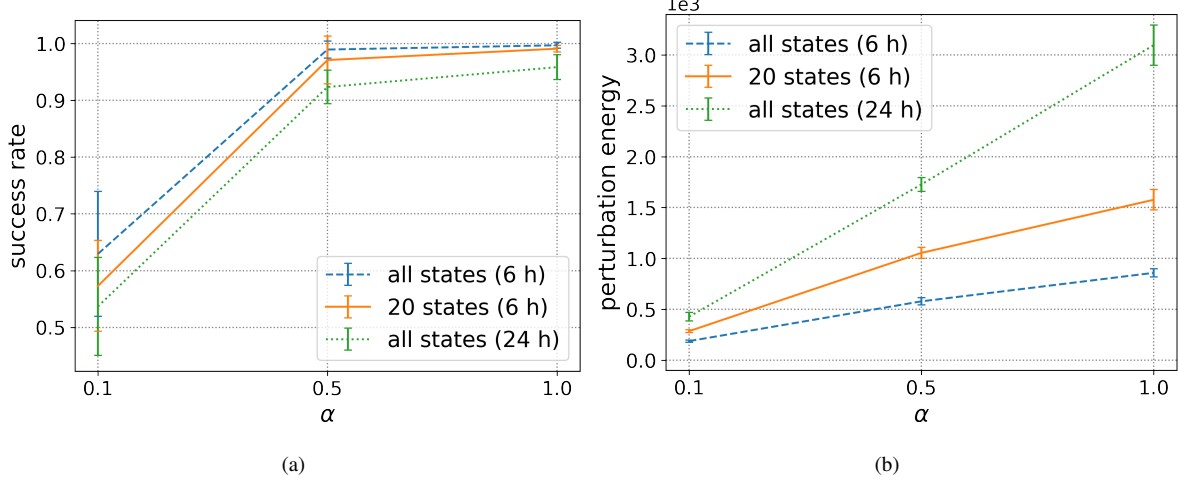

**Figure 11.** Success rate (a) and perturbation energy (b) for sparse observations experiments. The forecast time $T$ is set to 4 days.





## 5  Conclusions


In this study, we developed a CSE for reducing the occurrences of extreme events in the Lorenz-96 model with 40 variables. As background information, the main properties of the model are recalled, and an introduction to the LETKF is provided. The use of these tools for investigating the Lorenz-96 model has then been explained, and some parameters have been evaluated with preliminary experiments. Subsequently, we defined the notion of extreme events, and designed the experiment for reducing their

occurrence during periods of 100 years. The notions of success rate and of perturbation energy are introduced for discussing the outcomes of the experiments. Sensitivity tests on localized perturbations (vs global perturbations) show the potential of refining the experiments for a decay of the perturbation energy without a concomitant decay of the success rate. Additional sensitivity tests with partial observations show the robustness of the strategy. This study is an extension of previous investigations of controllability of chaotic dynamical systems based on the Lorenz-63 model (Miyoshi and Sun, 2022). Further investigations

on more complicated real world models or on numerical weather prediction models are planned. In this context, the potential side effects of the method should be carefully studied, and an global assessment of the CSE would be challenging but very interesting.

*Code availability.*  The code that supports the findings of this study is available from the corresponding author upon reasonable request.

*Author contributions.*  T. M. initiated the series of investigations about CSE and provided the fundamental ideas for controlling the extremes.

T. M. and S. R. advised the research. Q. S. and S. R. prepared the manuscript with comments from T. M.. Q. S. performed numerical experiments and visualized the results.

*Competing interests.*  T. M. is a member of the editorial board of journal Nonlinear Processes in Geophysics. The peer-review process was guided by an independent editor, and the authors have also no other competing interests to declare.

*Acknowledgements.*  T. M. thanks Dr. Ross Hoffman for informing him about his pioneering research work on weather control and on the

control of hurricanes. S. R. is supported by JSPS Grant-in-Aid for scientific research C no 18K03328 & 21K03292, and on leave of absence from Univ. Lyon, Université Claude Bernard Lyon 1, CNRS UMR 5208, Institut Camille Jordan, 43 blvd. du 11 novembre 1918, F-69622 Villeurbanne cedex, France. Q. S. is supported by RIKEN Junior Research Associate Program. This research work was partially supported by RIKEN Pioneering Project "Prediction for Science".





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
