# Peer review of "Control Simulation Experiments of Extreme Events with the Lorenz-96 Model"

_Nonlinear Processes in Geophysics, 2022_

## Author Response (AR2)

*First of all, let us thank the referees for their careful work on our manuscript. Their remarks allow us to improve the quality of the manuscript, and to clarify some parts of it. We list below the comments, and the corresponding changes in the manuscript.*

**Comments by referee 1:**

**One key aspect of control is the energy to be introduced in the system. The authors have here computed such an energy in the context of the Lorenz system, but it would be important to give a first clue to what quantity of energy would be necessary in a more realistic setting. As the Lorenz-96 model provides a toy model of the large scale variables at a specific latitude, it would be very interesting to convert the energy needed in an energy that the meteorological community could apprehend (power, work...) and discuss that in the conclusions.**

*This remark about energy is indeed a very important issue, and probably the next step in the elaboration of realistic CSEs. For our investigations, we have defined a notion of "energy" in order to quantify and compare the role of various parameters. However, due to the simplicity of the model, it is not clear if any valuable information can be extracted from this quantity. We looked at the literature if a proper notion of energy was elaborated and discussed for the Lorenz-96 model, but we could not find any reasonable one. As a consequence, we decided not to interpret our notion of energy, and to keep such a quantitative analysis for a more realistic model. Nevertheless, a comment about the notion of energy has been added in the conclusion.*
changes in manuscript: L319 ~ L337

**When perturbing a system (as done for instance with the increase of CO2), there are extremes that become less frequent like for instance a reduction of cold waves in certain regions with the increase of the global temperature. But this has other effects with an increase of heat waves. If one transposes this to the current setting, some extremes are suppressed, but some others might be arising. Did you see such type of**

**situations in the context of the Lorenz model? In any case it is necessary to elaborate on this somewhere in the manuscript.**

*This remark is interesting, and one related comment has been added in the conclusion of the manuscript. The Lorenz-96 model is probably too simple to clearly observe any side effects of reducing the positive extreme values. However, we thought about this and checked this effect with the following procedure. At the beginning of the investigations, it was decided to concentrate on the positive extreme values, disregarding the negative extreme values. Once the CSEs were performed, we checked if the statistics of the extreme negative values had changed, since this could have been a negative side effect of our control. It turns out that avoiding positive extreme values had statistically no impact on the negative ones. In fact, Figure 5 seems to indicate a small reduction of the negative extreme values, but similar figures for different parameters $\alpha$ and T did not show any significant changes. For that reason, we had decided not to report on this issue. With a more elaborate model, it is certainly an effect which should be carefully appraised.*

**In Figure 9, the authors show a saturation of the number of actions as a function of the localization scale. I am wondering whether it is related to the spatial correlation of the perturbations needed. Furthermore I am wondering what is the nature of the global perturbations. Do they look like bred modes? It would be really interesting to elaborate on that aspect.**

*Yes, in Figure 9(a), the saturation of the dashed line (neighbor sites) clearly indicates that perturbing additional sites which are far away from the site of interest does not lead to any benefit. This fact is indeed related to spatial correlations, since perturbations performed at random sites do not show such a saturation. This localization effect and this saturation are interesting for future applications: It shows that a local control is sufficient, with a corresponding reduction of the energy provided to the system, as illustrated in Figure 9(b).*

*About bred vectors, let us recall that the perturbation we identify is the difference between the EnKF ensemble state evolving to the extreme and the EnKF ensemble state evolving to the most non-extreme. The EnKF ensemble perturbations are considered as an extension of bred vectors. Therefore, the difference between the two ensemble states is somewhat related to the bred vectors. However, in our investigations, we did not elaborate further on bred vectors, but indeed in future studies we should look more carefully in this direction. In terms of effectiveness, working with bred vectors would allow us to provide the smallest perturbation to the system for the biggest impact.*

**The last paragraph of the introduction should be placed in the conclusion.**

*This paragraph has been removed and integrated in the conclusion.*
changes in manuscript: L319 ~ L337

**Line 126. There is no Appendix in the document.**

*This sentence has been removed.*
changes in manuscript: L119

**Comments by referee 2:**

**The manuscript is clear, the technical details of the experiments performed are well described. The authors provide several references to contextualize their research. This manuscript can be of interest for NPG readers, however my main concern is about the implications of this study in a more realistic context. The Control Simulation Experiment is aimed at reducing the extremes, but the challenge is that the models are able to simulate these extremes and in case of ensemble forecasting how the ensemble should be designed to include extremes. Therefore, I do not see the benefit of reducing the extremes in a simulation, when those states actually take place in the system that the model represents. I suggest including some clarifications in the introduction in this regard**

**as well as in the conclusions to better express the general objective of this research.**

*We agree that if the model can not predict extremes, our approach can not work. For this reason, our CSE is based on perfect-model assumption. The EnKF OSSE is designed that way, and the CSE is also designed with this assumption. Let us emphasize that current ensemble prediction systems tend to predict extreme events although not perfect yet. So, if we have good ensemble prediction systems in the future ,and at least one ensemble member showing the occurrence of extremes, we could potentially apply our CSE. Note that this underlying assumption has been added in the conclusion of our manuscript.*
changes in manuscript: L319 ~ L337

**L11: "of the first two authors" can be removed.**

*This has been corrected.*
changes in manuscript: L11

**L34-35: In line with my general comment, what is the benefit of reducing simulated weather extremes that occur in reality?**

*This has been discussed above.*

**L149: (j,j) -> (i,j)**

*This has been corrected.*
changes in manuscript: L140 ~ L144

**L170: Why only the maximum value is used to define extremes and not the minimum?**

*At the beginning of the investigations, this had been an arbitrary choice. Later on, it has been used to check if avoiding maximum values would generate the negative impact of having more negative values. As mentioned in a previous response, the number of extreme negative values has not changed statistically. In future investigations, avoiding both extreme values could be implemented.*

**L199: When the procedure to generate the perturbation vectors is described and the selection of the ensemble member B is explained, the alternative computation in case an ensemble member B is not found (L208-209) should be indicated here.**

*The alternative process has been moved to L192.*
changes in manuscript: L192 ~ L194

**L281: It is more correct saying similar or approximately equal instead of equal.**

*This has been corrected.*
changes in manuscript: L273

**L292-293: The meaning of this sentence is not clear.**

*The sentence has been moved to the subsequent paragraph, with additional explanation. In L284, the sentence has also been modified as: ….introduced in Sect. 2.3, need to be adapted for achieving the smallest RMSE.*

*The following sentence will be added at the end of the subsequent paragraph:*
*In Figure 10, the triangle indicates which combination of values for L and ρ leads to the smallest RMSE. This information is also reported in the second figure. By looking at the position of the two triangles, we observe that the RMSE and the average spread take approximatively the same value (about 0.31). Since the spread is computed as the RMSE but with the truth replaced by the mean value of the ensemble members, this concurrence means that the ensemble members spread sufficiently to cover the unknown truth of the model.*

changes in manuscript: L284 ~ L288

**Fig.10: Please, indicate the meaning of the triangle in the caption.**

*The following sentence has been added in the caption of Figure 10:*

*The triangles indicate which combination of values for L and ρ leads to the smallest RMSE. This information is also reported in the second figure.*

changes in manuscript: caption of Figure 10

*In addition, New Section 5 (conclusion) to be implemented.*

changes in manuscript: L310 ~ L338